# Effects of Chinese Martial Arts on Motor Skills in Children between 5 and 6 Years of Age: A Randomized Controlled Trial

**DOI:** 10.3390/ijerph191610204

**Published:** 2022-08-17

**Authors:** Bin Li, Ruijie Li, Haiquan Qin, Tao Chen, Jingyu Sun

**Affiliations:** 1Department of Physical Education, Tongji University, Shanghai 200092, China; 2School of Martial Arts, Shanghai University of Sport, Shanghai 200438, China

**Keywords:** Chinese martial arts, martial arts sensory teaching, martial arts traditional teaching, preschool children, motor skills

## Abstract

Children’s motor skills can be fully developed only by the appropriate stimulation of physical activities and the environment, and the poor development of motor skills greatly increases the risk of cognitive impairment, obesity, and movement coordination disorder. This study aimed to examine the effects of Chinese martial arts on the motor skills of preschool children aged 5–6 years through a randomized controlled trial. A total of 87 children aged 5–6 years served as participants in a martial arts sensory teaching group (MAST, *n* = 29), a martial arts traditional teaching group (MATT, *n* = 29), and a free activity group (FA, *n* = 29). The interventions were conducted twice weekly for a total of 10 weeks, with each session lasting 30 min. Children’s motor skills were assessed before and after the intervention using the Movement Assessment Battery for Children (MABC-2). The results indicated that the balance index scores in the MAST (*p* < 0.001) and MATT (*p* = 0.014) groups were significantly higher than those in the FA and that the MAST score was significantly higher than the MATT (*p* = 0.004). Meanwhile, the MAST was significantly higher in total scores on motor skills when compared to the FA (*p* = 0.039), and the MAST showed significantly higher scores on manual dexterity when compared to both the MATT (*p* = 0.021) and FA (*p* = 0.011). Chinese martial arts can significantly improve the balance ability of preschool children, and the MAST method was found to be better than that of the MATT. Meanwhile, the MAST had good potential for the development of preschool children’s manual dexterity and their overall level of motor skills.

## 1. Introduction

Motor skills refer to the ability of the human body to perform a variety of physical activities with specific components, including the fine motor skills of the hands (e.g., manual dexterity) and the gross motor skills needed for general coordination (e.g., static and dynamic balance) [1,2]. According to the mountain theory of motor skills development, the period between 1 and 12 years of age constitutes a sensitive time for the development of motor skills; children must master the fundamental motor skills, thus ensuring their future adaptation to different sports and movement environments while increasing their overall confidence in movement and enhancing their willingness to engage in physical exercise [3]. Further research also pointed out that the phenomenon of motor skills dysplasia in children has become increasingly prominent [4], and the incidence of motor coordination disorders in preschool children aged 3–6 years is as high as 5–7% [2]. Dysplasia of children’s motor skills not only greatly increases the risk of obesity, cognitive dysfunction, and motor coordination disorders [5,6], but also affects the development of children’s mental health. Children with motor coordination disorders often have poor social inclusion and lack confidence, and their anxiety and depression levels are often higher than those of normal children [7,8]. Therefore, the research on how to effectively improve the development of preschool children’s motor skills will be of great practical significance in promoting their healthy development.

The conceptual model theory of motor skills and physical activity development holds that there is a two-way interaction between motor skills and physical activity [9,10]. Structured physical activity intervention plays a positive role in promoting motor skills, and sports with diversified movement structures, collective exercise forms, and aerobic exercise with multi-cognitive participation are more conducive to the development of motor skills [10,11]. Martial arts, as an aerobic exercise requiring multi-cognitive participation and a diverse movement structure, has been gradually applied to the research on the development of motor skills in recent years. Multiple studies have been carried out in the field of judo, where the positive impacts of judo training have been shown to impact increased physical abilities of children when compared to other sports or inactive children [12,13,14,15]. Similarly, karate-related research has also obtained near results. Relevant studies have pointed out that karate has a good value in promoting coordination, strength, and flexibility in adolescents and children [16,17,18]. Physical fitness (strength, speed, endurance) is one of the factors that affect the performance of motor skills [19]. The studies above have paid attention to the impact of martial arts on the physical fitness of school-aged children. However, Taekwondo has been carried out in-depth research on the development of children’s motor ability. For example, Fong et al. found that the application of Taekwondo in children’s motor skill intervention experiments can effectively improve children’s balance ability [20]. At the same time, Shirley et al. conducted a 3-month Taekwondo teaching intervention for children with motor coordination disorders and found that Taekwondo training significantly improved the static balance ability of children with motor coordination disorders [21]. On the contrary, Ada et al. applied adaptive training methods and conducted intervention experiments with children with developmental coordination disorders through Taekwondo. The results showed that Taekwondo intervention did not significantly improve the balance and manual dexterity of children with developmental coordination disorders [22]. Different teaching intervention methods may affect the intervention. Taekwondo is a sport focusing on leg exercises but relatively little upper body training. Sufficient single-leg supports continuously stimulate the vestibule, develop the sensory function of the vestibule, and, thus, promote balance ability development [20]. The upper limb movements are mainly defensive, and the relative lack of muscle stimulation may be the main reason why the Taekwondo teaching intervention caused no significant improvement on children’s manual dexterity.

Different styles of martial arts have different sports characteristics and performance styles. Chinese martial arts is a traditional national sport comprising fighting methods and the main forms of routines, fighting, and skill [23]. In contrast to Taekwondo, Chinese martial arts not only focuses on the practice of leg technique, but also emphasizes the comprehensive application of a body technique, hand technique, and leg technique. Stodden pointed out that sensory ability and physical fitness are the mediating variables in the interaction between motor skills and physical activity [9,10]. Therefore, it has been speculated that highlighting the development of children’s sensory function and physical fitness in the instructional design of martial arts may be more conducive to the development of motor skills. However, at present, there is no report on the effects of different teaching intervention methods of Chinese martial arts on the motor skills of preschool children. According to the theory of embodied learning, the development of cognition depends on the environment, the body, and the interaction between the body and the environment [24]. Therefore, this study attempted to construct the sensory teaching method of Chinese martial arts under the guidance of embodied learning theory. The martial arts sensory teaching method creates a novel, warm, and rich martial arts activity environment, allowing children to interact in multiple dimensions with martial arts, equipment, companions, teachers, stories, etc., in the process of contextualization. In the process of multi-modal perception, they participate in gamified martial arts activities for an interactive experience of martial arts teaching methods. Relevant studies have shown that rich sensory stimulation can boost children’s learning motivation, immersing them in learning tasks, inducing positive psychological cues, and adjusting psychological states, thereby improving sensory function and developing physical fitness [25,26,27]. Based on the above research, we speculated that the application of a sensory teaching method to martial arts lessons in kindergarten may promote the motor skills development of preschool children. Therefore, this study used a randomized controlled trial to explore the effects of three different intervention methods (martial arts sensory teaching group (MAST), martial arts traditional teaching group (MATT), and free activity group (FA)) on the motor skills of preschool children. After 10 weeks, the effects of the three different intervention methods on the motor skills of 5–6-year-old preschool children were compared to explore the influence of Chinese martial arts and different teaching methods on the motor skills of preschool children. There were two hypotheses: (1) martial arts teaching interventions are better than free activities for promoting motor skills in preschool children; and (2) the MAST method is better than the MATT method for promoting motor skills in preschool children.

## 2. Materials and Methods

### 2.1. Participants

The sample size was calculated by G*Power 3.1 [28], indicating that 80 participants would be sufficient for 80% power (a = 0.05, effect size = 0.40) to test the primary outcome [29,30]. A total of 91 children aged 5–6 years in the kindergarten affiliated with Tongji University were selected as experimental subjects. Inclusion criteria: kindergarten children aged 5–6 years with parental consent and signed informed consent were included in the experimental study. Exclusion criteria: four children were excluded from the sample because two of them were, respectively, diagnosed with chronic physical diseases and autism by their doctors, and another two were absent from 50% of the experiment. Thus, the sample finally comprised 87 children, including 45 males and 42 females (average age: 5.5 ± 0.2 years). The subjects who met the inclusion criteria were randomly assigned to the MAST group, the MATT group, or the FA group in a proportion of 1:1:1. The randomization procedure was performed by an independent non-experimental design professional, and after the randomization sequence was generated by Excel software, a sealed opaque envelope was used for randomization. In the process of experimental intervention, double-blindness was in effect as often as possible. Except for the researcher himself, the experimental subjects and testers did not know the grouping information. After grouping, there were no significant differences in height, weight, age, and BMI index among the three groups, as shown in Table 1. Before the experiment, parents were informed of the research purpose and signed the informed consent. The subjects were children with similar social backgrounds, family incomes, and parent’s education levels, and their parents are staff of Tongji University. This study met the moral standards of the Helsinki Declaration and was approved by the Ethics Committee of Tongji University under the code: 2021tjdx024.

### 2.2. Experimental Treatment Variables

We selected “group” variables (MAST, MATT, FA) and “experimental time” variables (pre-experimental, post-experimental) as experimental processing variables and classification variables in the experimental design. To control the experimental errors caused by different teaching styles of other forms of physical education, experimental teachers were required not to disturb the original teaching plan as far as possible in the teaching of the experimental group and the control group. Except for guiding and helping the students of the two experimental groups in the martial arts teaching intervention, other teacher aspects were consistent in the experimental group and the FA group. To reduce the experimental errors caused by the teaching styles and abilities of different teachers, the MAST group, MATT group, and FA group were all taught by the same PE teacher.

### 2.3. Test Index and Test Methods

#### 2.3.1. Motor Skill

The Movement Assessment Battery for Children (MABC-2) was used to evaluate motor skills. There were three age bands that have specific scoring and task variations: age band 1 (3–6 years old), Age band 2 (7–10 years old), and Age band 3 (11–16 years old). The tasks of Age band 1 were used for the present study [31], and the MABC-2 was widely used to evaluate motor skills in young children [32]. There were three movement dimensions: manual dexterity, aiming and catching, and balance (eight total test contents). Manual dexterity was based on the completion times for three activities: placing coins, stringing beads, and drawing paths. Aiming and catching were based on the number of successful hits and catches in two activities: for aiming, participants successively threw 10 bean bags measuring 12 × 12 cm at a distance of 1.8 m from a target mat; for catching, participants were required to successively catch the same bean bags as thrown from a distance of 1.8 m. Balance was based on three activities: one-legged standing, walking on tiptoe in a straight line, and two-legged continuous jumping. For one-legged standing, participants stood still on their left or right leg, with the respective lengths of time recorded. For walking on tiptoe, participants slightly lifted their heels while walking a straight line measuring 4.5 m in length and 0.25 m in width, with the number of steps recorded. For two-legged continuous jumping (5 times), participants used both legs to jump on six mats of different colors in sequence, with the number of successful attempts recorded [22,33].

The administration of the MABC-2 was divided into six stations: two stations for manual dexterity, two stations for aiming and catching, and two stations for balance. During each session of data collection, approximately 10 to 12 children were divided among the six stations. Children rotated from station to station, with the aid of research assistants, until the entire MABC-2 was completed. There were eight trained researchers, five faculty members, and three graduate students in the area of Motor Behavior who administered the MABC-2. The testing time was 8:00–11:30, in a ventilated, bright and quiet gymnasium. Before the test, the staff first demonstrated and explained the action method. Except for aiming and catching, which were practiced 5 times, other items were practiced once, and the test was carried out after the practice. Each researcher had prior experience in administering motor assessments in both research and practical settings. One to two researchers were assigned to each station to ensure accurate scoring of the tasks. All researchers attended a 2 h training session to become familiar with the assessment procedures. Scoring was based on quantitative measures, such as the number of successful trials and the length of time taken to complete a task. After recording raw data for each test, the scores were entered into the MABC-2 testing system, thus resulting in raw, standard, and percentile scores for the total scores on manual dexterity, aiming and catching, and balance. This study statistically analyzed the raw scores from all three groups. The MABC-2 had retest reliability of 0.75 and validity of 0.70 [1], with good internal consistency (Cronbach’s α range of 0.81–0.90); the total scale and both its sub-items and sub-scales showed good retest reliability [2].

#### 2.3.2. Sensory Integration Ability

Participants were also subjected to the Sensory Integration Ability Development Assessment Scale for Children, as developed by Professor XinXiong Zheng and revised by Yufeng Wang and Guiying Ren [34]. The scale consisted of five items, including vestibular balance dysfunction, tactile over-defense, proprioceptive dysfunction, insufficient learning ability, and special problems of older age, with all items rated on a 5-point scale (1 = never, 2 = rarely, 3 = sometimes, 4 = often, 5 = always). Insufficient learning ability and special problems of older age were excluded from this study for being designed for school-aged children. The raw scores for each scale item were converted into standard scores, with the norm provided by the Institute of Mental Health, Beijing Medical University. Total scores of 50 or higher represented normal functioning, while scores below 40 represented mild sensory integration disorder, and scores below 30 represented severe sensory integration disorder [35]. The scale had good reliability and validity, with retest reliability of 0.47–0.73, split-half reliability of 0.68–0.7, homogeneity reliability of 0.44–0.63, and validity of 0.49–0.94 [34]. One week before the baseline test and one week after the intervention, paper evaluation scales were distributed, respectively, and the guardians filled in after the children brought them home, and collected them the next day. The recovery rate was 100%.

### 2.4. Experimental Procedure

First, the relevant administrators and teachers of the kindergarten were contacted to explain the objectives of the study, and with the permission of the kindergarten, the informed consent form for the study was sent to the parents of the children, signed, and returned.

Secondly, basic information and measurements were obtained for all participants (height, weight, date of birth, gender, and physical health status). Participants were excluded in cases where informed consent could not be obtained or when they had learning difficulties and/or behavioral, neurological, or orthopedic problems.

Third, trained and qualified assessors use standardized MABC-2 testing tools to test children’s motor skills in a quiet classroom that was bright, clean, and well-ventilated. Before taking the test, the assessor first explained the test method, the children could practice twice, and the assessor guided the children in the practice. The test was conducted immediately after the practice session, and the child was not instructed during the test.

Fourth, the two martial arts teaching groups, respectively, conducted the martial arts sensory teaching intervention and martial arts traditional teaching intervention. The FA group conducted outdoor free activities according to the school plan.

Fifth, after the 10-week intervention, the children’s motor skills were retested by raters blinded to the assignments. One week before and after the intervention, sensory integration tests were performed on the three groups of children, and the test was completed by the test subject’s guardian based on the test subject’s situation within the past month.

### 2.5. Intervention

The MAST group and the MATT group practiced martial arts for children. These contents were edited by the State Sports General Administration [36]. The practice lasted for 30 min per time twice a week for 10 weeks. The Chinese children’s martial arts movements were choreographed and coordinated as a whole. The content of martial arts is Changquan. The coordination of upper and lower limbs, coordination of limbs and body, coordination of visual movements, and hand movements were considered in the criteria for content selection. Detailed elements, sequences, and training plans are presented in the Appendix A. In contrast, the FA group merely engaged in freely chosen outdoor activities (e.g., sliding, swinging, and bicycling). In terms of teaching methods, the MAST group aimed at the children’s active participation in the experience, emphasizing participation in gamified martial arts activities in a rich environment and stimulating body senses, to obtain a rich embodied experience; while the MATT group had the goal of mastering martial arts skills, and the movement quality of martial arts was optimized in repeated intensive exercises; the free activity group was mainly based on self-selected activities. The teaching design method of each group is shown in Table 2. Exercise intensity was gauged via heart rate monitors (Finland, polar RS800cx) that were randomly provided to 10 participants in each group, with measurements taken at 5 min intervals to ensure exercising heart rates of 120–160 beats per minute [37].

### 2.6. Statistical Methods

The data were imported from Excel to the SPSS23.0 version of the dataset (IBM, Armonk, New York, NY, USA) for descriptive and inferential statistical analysis to screen outliers and non-normal data before analysis. First, the Shapiro–Wilk test was employed to inspect the normality and homogeneity of variance of all the data. The basic characteristics of the sample were statistically analyzed through descriptive analysis, including gender and the average and standard deviation of all variables. Secondly, the multivariate analysis of variance of repeated measurement was used to examine the changes in motor skills performance (manual dexterity, aiming and catching, balance, and total score) before and after the experimental intervention. The inter-group factor was the group (MAST, MATT, FA), while the intra-group factor was time. At the same time, taking gender and group as inter-group factors and time as intra-group factors, a repeated-measures multivariate analysis of variance was conducted to verify whether the intervention effect had gender differences. Any results not satisfied through the sphericity test were adjusted using the Greenhouse–Geisser method, while the Bonferroni method was used to correct multiple comparisons post-test (significance at 0.05 for all analyses).

## 3. Results

### 3.1. Motor Skills

#### 3.1.1. Total Motor Skills Score

A repeated-measures ANOVA on the total motor skills scores showed significant main effects between both the pre-test and post-test results (*F*(1,81) = 20.41, *p* < 0.001, *η_p_*^2^ = 0.201) and groups (*F*(2,81) = 3.350, *p* = 0.04, *η_p_*^2^ = 0.076). There were no significant interaction effects between pre-test/post-test results and groups (*F*(2,81) = 2.353, *p* = 0.102, *η_p_*^2^ = 0.055). Before the intervention, there was no statistical difference in the total motor skills scores among the three groups (*F*(2,84) = 1.045, *p* = 0.35), while after the intervention the total motor skills scores increased in all three groups (Table 3). Both the MAST (*p* < 0.001) and MATT (*p* = 0.026) groups had significantly higher post-test scores when compared to their respective pre-test scores; this difference was not significant for the FA (*p* = 0.237). Although a post hoc analysis showed that the MAST group had significantly higher scores than the FA group (*p* = 0.039), these differences were not significant between the MAST and MATT (*p* = 0.671) or MATT and FA (*p* = 0.573) groups (Figure 1).

#### 3.1.2. Manual Dexterity

A repeated-measures ANOVA on the manual dexterity scores showed that the interaction effect between pre-test/post-test results and group (*F*(2,81) = 1.421, *p* = 0.247, *η_p_*^2^ = 0.034) was not significant. However, significant main effects were found for both pre-test/post-test results (*F*(1,81) = 8.35, *p* = 0.005, *η_p_*^2^ = 0.093) and group (*F*(2,81) = 5.912, *p* = 0.004, *η_p_*^2^ = 0.127). Before the intervention, there was no statistical difference in the manual dexterity among the three groups (*F*(2,84) = 1.806, *p* = 0.171), while all three groups showed improved manual dexterity after the intervention period (Table 3). The paired sample T-test results showed that the post-test scores of the MAST group were significantly higher than the pre-test scores (*p* = 0.003), while there were no significant differences between the pre-and post-test scores of the MATT (*p* = 0.249) and FA (*p* = 0.368) groups. A post hoc analysis showed that the MAST group had significantly higher scores than both the MATT (*p* = 0.021) and FA (*p* = 0.011) groups; scores were not significantly different between the MATT and FA groups (*p* = 1.000) (Figure 1).

#### 3.1.3. Aiming and Catching

A repeated-measures ANOVA on the aiming and catching scores showed a significant main effect between the pre-test and post-test results (*F*(1,81) = 5.315, *p* = 0.024, *η_p_*^2^ = 0.062), but no significant differences between groups (*F*(2,81) = 2.356, *p* = 0.101, *η_p_*^2^ = 0.055). There were no significant differences in the interaction effects (*F*(2,81) = 0.057, *p* = 0.945, *η_p_*^2^ = 0.001) between groups and pre-test/post-test results (*F*(2,81) = 0.072, *p* = 0.931, *η_p_*^2^ = 0.002).

#### 3.1.4. Balance

A repeated-measures ANOVA on the balance scores showed significant main effects between both the pre-test and post-test results (*F*(1,81) = 47.44, *p* < 0.001, *η_p_*^2^ = 0.369) and groups (*F*(2,81) = 9.512, *p* < 0.001, *η_p_*^2^ = 0.190). There were significant differences in the interaction effects between pre-test/post-test results and groups (*F*(2,81) = 11.20, *p* < 0.001, *η_p_*^2^ = 0.217). The simple effects analysis showed no significant intergroup differences in balance scores before the intervention. Post-intervention, the MAST group showed significantly higher scores than both the FA (*p* < 0.001) and MATT (*p* = 0.004) groups, while the MATT showed a significantly higher score than the FA (*p* = 0.014) group (Figure 1). The paired samples T-test indicated that both the MAST (*p* < 0.001) and MATT (*p* = 0.001) groups had significantly higher post-test scores when compared to their respective pre-test scores; this difference was not significant for the FA group (*p* = 0.453) (Table 3).

Repeated-measures ANOVA results showed that there were no statistical differences in gender in the total motor skills scores (*F*(2,81) = 1.867, *p* = 0.18, *η_p_*^2^ = 0.023), manual dexterity scores (*F*(1,81) = 1.042, *p* = 0.310, *η_p_*^2^ = 0.013), aiming and catching scores (*F*(1,81) = 1.743, *p* = 0.190, *η_p_*^2^ = 0.021), and balance scores (*F*(1,81) = 0.044, *p* = 0.834, *η_p_*^2^ = 0.001).

### 3.2. Sensory Integration

#### 3.2.1. Vestibular Function

A repeated-measures ANOVA on the raw vestibular function scores showed a significant main effect for pre-test and post-test results (*F*(1,84) = 8.53, *p* = 0.004, *η_p_*^2^ = 0.09), a non-significant main effect for groups (*F*(2,84) = 2.13, *p* = 0.125, *η_p_*^2^ = 0.05), and significant interaction effects between groups and pre-test/post-test results (*F*(2,84) = 11.18, *p*
*<* 0.001, *η_p_*^2^ = 0.21). Pre-intervention, a simple effects analysis showed that the MAST group had significantly lower vestibular function scores than the FA group (*p* = 0.005), but that such differences were not significant between the MATT and FA (*p* = 0.41) or MAST and MATT (*p* = 0.25) groups. Post-intervention, there were no significant intergroup differences in these scores (*p* < 0.05). As for intragroup comparisons, the MAST group had significantly higher vestibular function scores after the intervention (*p* < 0.001), while the FA group showed slightly decreased scores (*p* = 0.39) (Table 4).

#### 3.2.2. Tactile Function

A repeated-measures ANOVA on the raw tactile function scores showed a significant main effect for the pre-test and post-test results (*F*(1,84) = 27.89, *p* < 0.001, *η_p_*^2^ = 0.25), a non-significant main effect for group (*F*(2,84) = 0.88, *p* = 0.42, *η_p_*^2^ = 0.02), and significant interaction effects between groups and pre-test/post-test results (*F*(2,84) = 11.49, *p*
*<* 0.001, *η_p_*^2^ = 0.22). Pre-intervention, a simple effects analysis showed that the MAST group had significantly lower tactile function scores than the FA group (*p* = 0.04), but no such differences were found between the MATT and FA (*p* = 0.26) or MAST and MATT (*p* = 1) groups. Post-intervention, there were no significant intergroup differences in tactile function scores. For intragroup comparisons, both the MAST (*p* < 0.001) and MATT (*p* = 0.012) groups had significantly higher post-test tactile function scores when compared to their respective pre-test scores; the FA group had slightly lower post-test scores (*p* = 0.942) (Table 4).

#### 3.2.3. Proprioception

A repeated-measures ANOVA on the raw proprioception scores showed significant main effects for both the pre-test and post-test results (*F*(1,84) = 25.22, *p* < 0.001, *η_p_*^2^ = 0.23) and group (*F*(2,84) = 3.26, *p* = 0.04, *η_p_*^2^ = 0.07), as well as significant interaction effects between group and pre-test/post-test results (*F*(2,84) = 16.48, *p* < 0.001, *η_p_*^2^ = 0.28). Pre-intervention, a simple effects analysis showed that the MAST group had significantly lower proprioception scores than the FA group (*p* < 0.001), while the MATT group had significantly higher scores than the MAST group (*p* = 0.02), but no such differences were found between the MATT and FA groups (*p* = 0.44). Post-intervention, there were no significant intergroup differences in proprioception scores. For intragroup comparisons, the MAST group had significantly higher post-test proprioception scores when compared to their pre-test scores (*p* < 0.001); there were no such differences for either the MATT (*p* = 0.09) or FA (*p* = 0.65) groups (Table 4).

## 4. Discussion

After 10 weeks, at 30 min twice a week, the martial arts teaching intervention groups (MAST and MATT) scored significantly better than the FA group in the balance ability test index. In addition, we found that the MAST was better than the MATT in promoting the balance ability of preschool children, and the MAST improved the manual dexterity and total motor skills of preschool children. At the same time, there was no gender difference in the promoting effect of the martial arts teaching intervention on preschool children’s movement ability. The results of this study partially verified the expected hypothesis.

### 4.1. The Effects of Chinese Martial Arts on Balance Ability

Post-intervention, the MAST and MATT groups showed better balance than the FA group. This was consistent with previous research. For example, a systematic review showed that structured physical activity interventions could improve balance in children between 2 and 6 years of age [10]. An earlier study found a similarly significant positive correlation between balance ability and physical activities [11]. Balance is the ability of the human body to maintain body posture in dynamic or static conditions [38]. Postural adjustments and body balance require coordination of the vestibular, proprioceptive, and visual sensory organs, which transmit information on acceleration and spatial displacement to the central nervous system, which analyzes and integrates this information, then sends it through the vestibular spinal cord to the effector, thus regulating muscle tension, relaxation, and balance control [39,40]. Chinese martial arts have high requirements for posture control. During practice, the external hands, eyes, torso, and feet are required to move in harmony as a whole, and the inner spirit, will, breathing, and strength are highly unified [23,41]. “External training” requires the perfect coordination of body postures, such as hand, eye, body, and footwork. The hands and eyes are coordinated, and the trunk and footsteps are coordinated. “Internal training” requires the inner unity of mind, spirit, offense-defense conversion, and breathing, so that the breathing and force are coordinated, and the mind and breathing are coordinated [42,43]. The results of the sensory integration test of preschool children in this study demonstrated that Chinese martial arts can effectively improve the vestibular and proprioceptive functions of preschool children, as shown in Table 4. Compared with free activities, Chinese martial arts, which emphasizes the overall coordination of the external body and the high concentration of internal ideas, effectively promotes the development of children’s sensory functions, such as vestibule and proprioception. This may be why Chinese martial arts was better than FA in promoting the balance ability of preschool children. The results of Ada’s research also supported this conclusion. Martial arts athletes have better motor control and coordination ability of the upper limbs. It may be that the overall high coordination of movements promotes the development of martial arts athletes’ proprioceptive function and shortens the delay of the biological force produced by the upper limb muscle strength [14,16,22]. In addition, martial arts training can enhance explosive strength, and improve flexibility and reaction speed. It may also be an important factor for Chinese martial arts to promote preschool children’s balance ability [12,13,15].

The theory of motor development sequence holds that the development of children’s motor skills follows a certain sequence, and the mastery of each motor skill needs to undergo the construction process of repeated attempts, adjustment, and adaptation [44]. Therefore, professional guidance and repeated practice are essential for the development of children’s motor skills [45]. Repeated practice is the basic means of Chinese martial arts training; each action must be practiced repeatedly to achieve the basic mastery and then progress from mastering the action to the complete automation of the action. A previous study found that structured physical activity interventions that continuously reinforce motor skills produced significant improvements over free activities in children [9]. Therefore, compared with free activities, Chinese martial arts significantly improves the balance ability of preschool children, which may be related to the repeated intensive training in martial arts projects.

From childhood to early adolescence, there is a positive reciprocal relationship between physical activity and motor skills, with aerobic endurance bridging bidirectional development [46]. Research shows that purposeful, planned, and organized structural sports activities can create more opportunities for children to benefit from medium and high intensities of aerobic exercise [47]. A randomized controlled trial also showed that children who engage in planned structured physical activity spend significantly more time in moderate-to-high motor intensity activity when compared to free activity engagement [48]. Therefore, martial arts teaching intervention, as a purposeful, planned, and organized structured physical activity, may create more lasting opportunities for children to exercise at medium and high intensities and be more conducive to the development of balance ability.

### 4.2. The Influence of MAST on the Motor Skills of Preschool Children

MAST resulted in greater motor skills improvements than MATT. This may be due to the novel teaching environment and rich sensory stimulation. First, continuous interaction between the body, the environment, and the given task influence motor skills development [5,45]. This is supported by Bronfenbrenner’s ecosystem theory [9]. In children, postural control not only improves with age but also relies on effective environmental interaction and precise muscle conditioning through repeated exercise [39]. Motor and executive control are refined through continuous interactions between the body, task, and environment, with postural regulation becoming more flexible [10]. The MAST method integrates martial arts with stories, games, and competitions, thus promoting active participation in movement and exercise in an open, diverse, and free environment. For children, these teaching designs can motivate participation, increase movement intensity and exercise density, and promote interactions between the body, task, and environment [49]. Meanwhile, active practice improves brain concentration, which enhances cognitive processing speed and postural control [50,51]. Compared with MAST, MATT is mainly based on the collective and repetitive practice form, and the repetitive and passive practice method lacks effective interaction between the body, the environment, and the task. Studies have also found that suitable exercise patterns and learning environments can promote the sustainable development of children’s motor skills [9,52]. Therefore, the teaching design characteristics of paying attention to the creation of the environment and emphasizing the continuous interaction between the body, the task, and the environment may be the reason why MAST is better than MATT in promoting preschool children’s balance ability and manual dexterity.

Secondly, multimodal sensory stimulation can strengthen the function of sensory integration and is more conducive to the development of motor ability [53]. In sensory–motor integration, sensory input signals are integrated by the central nervous system to assist in the execution of motor programs. Sensory organs input sensory information to the brain, which synthesizes information according to the principle of maximum reliability and creates weightings through corresponding sensors to achieve effective perception and motor control [54]. The effective integration of sensory information is a necessary condition for the realization of action control [55]. Of note, children are self-referenced and goal-oriented. Integrated sensory information can describe the position, size, weight, shape, and displacement speed of objects, further optimizing action plans through continuous motor sensory feedback [55]. The simultaneous multi-sensory stimulation of the vestibule, muscles, joints, skin, and the visual, auditory, and olfactory senses forms multimodal information processing patterns and accelerates the formation of motor representations for better postural control [56,57]. Based on the principle of sensory integration, the MAST method aims for multi-sensory stimulation and increases constant interactions among the senses. Indeed, MAST was significantly better than MATT for promoting vestibular, tactile, and proprioceptive functions (Table 4); here, the enhancement of sensory integration skills may have exerted a stronger mediating effect on movement ability.

### 4.3. Effects of Martial Arts Teaching on Aiming and Catching According to Gender

In this study, martial arts instruction did not significantly promote aiming and catching, which was contrary to a previous structured physical activity intervention showing aiming and catching improvements in non-normally sighted children [53]. These inconsistencies may be related to the intervention contents, as the abovementioned study directly applied to aim and catching activities, which may be more likely to develop relevant motor skills through targeted reinforcement training. In this study, the interventions were based on unarmed exercises, which entail less reinforcement training for object control and grasping movements. This also suggested that it is very necessary to increase the practice with martial arts equipment (such as broadsword, stick, and sword) in the intervention content.

The results of the study also indicated that there was no gender difference in the effect of the martial arts teaching intervention on the motor skills of preschool children. This result was inconsistent with the conclusions of previous studies. Related studies have pointed out that after exercise intervention, both boys and girls benefit in motor ability scores, while boys benefit more in goal control skills, and girls improve faster in manual dexterity [58,59]. However, no gender difference was found in this study. In addition to the small sample size, different intervention contents and different teaching organization methods may also be important factors leading to the result. The content of martial arts training is relatively fixed, and the practice form is relatively standardized and orderly, so boys and girls can maintain a good consistency in exercise intensity and practice density. Therefore, the effect of the martial arts teaching intervention on the motor skills of preschool children had no gender difference, which may be related to the small sample size and the good consistency of exercise content, exercise intensity, and exercise density between male and female students.

## 5. Study Limitations and Future Research

This study investigated the influence of Chinese martial arts on motor skills in preschool children, with measurements taken via the MABC-2. All participants shared similar background characteristics, thus avoiding some external influences. However, there were some limitations. First, the short intervention period and small sample size limited generalizability. Second, as there was no long-term follow-up, it is unclear whether the effects are enduring. Future studies should expand the experimental scope by recruiting larger samples over longer periods. Researchers should also investigate different training frequencies and duration among different ages and genders. Third, an effective method to control the activity level can improve the stability and reliability of the results. To explore the activity level of children in and out of class, future studies should resort to questionnaires, interviews, or accelerometers to reduce the influence of irrelevant intervention factors on the experimental results. Therefore, future research designs could improve themselves in this regard.

## 6. Conclusions

Chinese martial arts can effectively promote the balance of children aged 5–6 years, and the effect of martial arts sensory teaching on the balance of preschool children is significantly better than that of traditional martial arts teaching. Chinese martial arts sensory teaching can potentially improve the manual dexterity and total motor skills score of children aged 5–6. Consequently, it is suggested that regular practice/use of Chinese martial arts should be included in preschool education settings, or rehabilitation institutions for children with motor developmental disorders.

## Figures and Tables

**Figure 1 ijerph-19-10204-f001:**
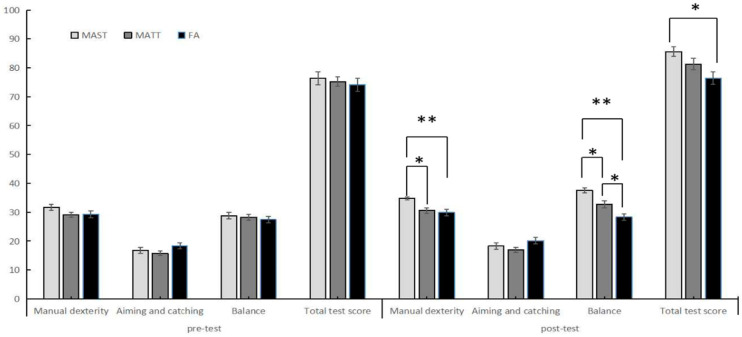
Pre- and post-test scores and standard errors for the motor ability test in all three study groups. * *p* < 0.05, ** *p* < 0.01.

**Table 1 ijerph-19-10204-t001:** Participant information.

	MAST (*n* = 29)	MATT (*n* = 29)	FA (*n* = 29)	*F*	*p*
Height (cm)	113.6 ± 5.3	116.8 ± 4.9	114.4 ± 6.1	2.76	0.069
Weight (kg)	20.9 ± 4.5	21.2 ± 2.8	21.7 ± 3.4	0.4	0.67
Age (year)	5.6 ± 0.2	5.4 ± 0.3	5.5 ± 0.3	1.33	0.27
BMI	16.1 ± 2.8	15.5 ± 1.4	16.5 ± 1.7	1.96	0.15

**Table 2 ijerph-19-10204-t002:** Comparison of the intervention processes (experimental groups and control group).

MAST	MATT	FA
(1) Learning situation analysis: analysis of learners, teaching objectives, teaching conditions, and teaching content (2) Structured martial arts content: reorganize the order of teaching content; present content reasonably; make content interesting and simple (3) Learning environment preparation: arrange novel and exciting activity spaces; set up vivid sports venues; create a warm learning atmosphere (4) Activity organization: stimulate learning motivation; construct teaching situation; game-based teaching activities to provide effective help (5) Interactive experience: multi-dimensional interaction; multi-modal body perception (6) Evaluation	(1) Learning situation analysis: analysis of teaching conditions and teaching content (2) Orderly teaching: according to the movement structure of martial arts routines, teaching in sequence from front to back (3) Combination of teaching new and consolidation: review before class, improve and consolidate the learned content, and learn new content after the review (4) Group exercises: after the new content is basically mastered, use group exercises to strengthen motor skill (5) Evaluation	(1) Learning situation analysis: analyze teaching conditions (2) Teacher supervision: participate in outdoor activities independently under the supervision of the teacher. The teacher is mainly responsible for the safety of children and does not guide activities

**Table 3 ijerph-19-10204-t003:** Movement ability results for all groups (mean ± sd).

Outcome Variables		MAST (*n* = 29)	MATT (*n* = 29)	FA (*n* = 29)	*p*		
					Group	Time	G × T
Manual dexterity	Pre	31.69 ± 5.83	29.14 ± 4.71	29.28 ± 6.53	0.004	0.003	0.224
	Post	34.76 ± 2.90	30.62 ± 5.07	29.93 ± 6.02			
Aiming and catching	Pre	16.86 ± 5.44	15.79 ± 4.18	18.41 ± 5.52	0.054	0.024	0.931
	Post	18.28 ± 5.84	16.97 ± 4.70	20.14 ± 6.28			
Balance	Pre	28.79 ± 5.70	28.28 ± 5.55	27.45 ± 5.99	<0.001	<0.001	<0.001
	Post	37.59 ± 4.44	32.69 ± 6.38	28.38 ± 5.98			
Total score	Pre	76.34 ± 11.86	75.21 ± 8.72	74.14 ± 11.83	0.04	0.001	0.102
	Post	85.62 ± 8.73	81.28 ± 10.29	76.45 ± 11.04			

**Table 4 ijerph-19-10204-t004:** Sensory integration results for all groups (mean ± sd).

		MAST (*n* = 29)	MATT (*n* = 29)	FA (*n* = 29)	*p*		
					Group	Time	G × T
Vestibular function	Pre	52.86 ± 9.08	57.07 ± 9.96	60.69 ± 8.33	0.125	0.004	<0.001
	Post	59.69 ± 6.17	57.62 ± 7.37	59.62 ± 7.24			
Tactile function	Pre	89.21 ± 11.89	91.48 ± 11.32	96.28 ± 7.79	0.42	<0.001	<0.001
	Post	98.69 ± 6.86	98.69 ± 6.68	96.17 ± 8.65			
Proprioception	Pre	51.07 ± 7.23	55.41 ± 6.00	57.62 ± 3.22	0.04	<0.001	<0.001
	Post	57.83 ± 4.03	57.00 ± 4.36	57.21 ± 4.59			

## Data Availability

Not applicable.

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
