# Peer review of "Effects of Chinese Martial Arts on Motor Skills in Children between 5 and 6 Years of Age: A Randomized Controlled Trial"

_ijerph, 2022, doi:10.3390/ijerph191610204_

Round 1
Reviewer 1 Report
Dear Authors
You have written an interesting paper studying the effects of Chinese martial arts on the motor skills of preschool children aged 5-6 years.
However, some parts of the manuscript need to be addressed for greater clarity and reproducibility.
Remove the words martial arts from keywords as it is already in the title.
Introduction:
The introduction is solid. However, you just present a case of taekwondo. You have a lot of studies from judo that show a positive impact on the motor development of children:
Drid, P., Ostojić, S., Maksimović, N., Pejčić, J.,.Matić, R. in Obadov, S. (2009). The effects of judo training on anthropometric characteristics and motor abilities of primary schoolboys. Homo sporticus, 1, 28–32.
Fukuda, D. H., Stout, J. R., Burris, P. M. in Fukuda, R. S. (2011). Judo for Children and Adolescents: Benefits of Combat Sports. Strength and Conditioning Journal.33(6), 60–63.
Krstulović, S., Kvesić, M. in Nurkić, M. (2010). Judo training is more effective in fitness development than recreational sports in 7-year-old girls. Facta Universitatis, 8(1), 71–79
Sekulić, D., Krstulović, S., Katić, R. in Ostojić, L.17.(2006). Judo training is more effective for fitness development than recreational sports for 7-year-old boys. Pediatric Exercise Science,18, 329–338
Here are just a couple of papers that report in favour of combat sports and martial arts. This could also help you strengthen your discussion. Therefore, try to incorporate them in your introduction and discussion and make a little wider review of the literature.
Line 108 - please explain the abbreviations MAST and MATT when used for the first time in the main text.
Participants: How was your sample size determined (G*Power software or any other method). Please report.
Line 116- abnormal intelligence? Please explain what was meant by this abnormality as it can be very high or very low.
Line 120 – why do you report age in months? Delete mean and sd words.
Report how were weight and height tested and equipment used.
Report activity levels for your participants as this can affect their starting level of motor abilities.
Motor skills: Line 152 – Please explain why did you use a testing battery that is used for children with motor disorders if your sample of children did not report any disabilities? From this point of view, children with normal development should perform very well in this kind of test and I don’t see a point in the results. Please elaborate
How many times was each test performed, what was measured, and which result was taken into further analysis needs to be reported for every test. Was there any demonstration prior to the test? Please explain.
Also report, at what time and where was this done. How many people evaluated this test?
Lines 170-172. Please explain how did you calculate reliability and validity? To which test was this battery compared for validity? Please report. Also, how did you calculate the reliability? From your description, it looks like you calculated it from data before and after the intervention. Also, report exactly how was internal consistency calculated (compared to what)? Elaborate and report
Sensory Integration Ability – please report when was this done and who evaluated your participants. Please describe those 5 items of the assessment.
Intervention – in this state your study is not reproducible. If could reproduce only the FA group with free activity. However, you need to report, how many times per week was this performed, and how long were these sessions.
Also, which Chinese Martial arts or styles were trained? Please report
How can I, from your description reproduce the MAST and MATT training? I can’t!
Therefore, please provide supplemental material describing exact elements that were thought and sequences with photo material, preferably! Also, provide information on what was thaught by weeks.
Statistical methods – how was the normality of data tested? Please report
Overall a very interesting and promising study. However, the methodological part is of poor quality as it affects the reproducibility of the current study. Especially the intervention part.
Therefore, I recommend a major revision.
Kind regards
Author Response
Dear Reviewer:
On behalf of my co-authors, we are very grateful to you for giving us an opportunity to revise our manuscript. We appreciate you very much for your positive and constructive comments and suggestions on our manuscript entitled “Effects of Chinese Martial Arts on Motor Skills in Children Between 5 and 6 Years of Age: A Randomized Controlled Trial”.
We have studied reviewers’ comments carefully and tried our best to revise our manuscript according to the comments. The following are the responses and revisions. I have made in response to the reviewers’ questions and suggestions on an item-by -item basis. Thanks again to the hard work of the editor and reviewer!
Comment No.1: Introduction:The introduction is solid. However, you just present a case of taekwondo. You have a lot of studies from judo that show a positive impact on the motor development of children:
Response: First of all, thank you very much for pointing out the shortcomings of my thesis in the foreword and discussion, and for collecting relevant literature for me. Your serious work attitude is what I need to learn in writing in the future. In the preface, we added the content of judo and karate to further explain the influence of martial arts on children's motor skill (because martial arts may gradually develop by improving children's physical fitness, such as strength, reaction speed, endurance, flexibility, etc. Gross and fine motor movements in children). We marked in red line from 59 to 71 in the revised article.
In the discussion part, we also inserted the content of judo and karate, because judo and karate can effectively develop children's leg strength and flexibility, and strength and flexibility are one of the factors that affect the balance of the body, so in the discussion part above content was added in the thesis to demonstrate the influence of Chinese martial arts on children's balance ability. Supplements are also marked in red from line 430 to 432.
Comment No.2: Line 108 - please explain the abbreviations MAST and MATT when used for the first time in the main text.
Response: Thanks to Reviewer for reminder, we have put the MAST, MATT and FA on the part of Introduction, from line 117 to 118.
Comment No.3: Participants: How was your sample size determined (G*Power software or any other method). Please report.
Response: The sample size was calculated by G*Power 3.1 (http://www.gpower.hhu.de/en.html), indicating that 80 participants would be sufficient for 80% power (a = 0.05, effect size = 0.40) to test the primary outcome. In the process of sample size measurement, we have consulted a large number of relevant literature, and we have determined the basic parameters of the G*Power 3.1 measurement of the sample in this study based on the references No. 26 and 27, so as to calculate the sample size required for this study. From lines 128 to 130, indicated in red.
Comment No.4: Line 116- abnormal intelligence? Please explain what was meant by this abnormality as it can be very high or very low.
Feedback: Actually he is a child with autism diagnosed by his doctor, his intelligence is below normal, and also be reported by his parents that he can’t do normal activities with his classmates. So he was excluded to our Intervention. We also have explained in details in the article from line 134 to 135.
Comment No.5: Line 120 – why do you report age in months? Delete mean and sd words.
Feedback: we have changed the age in years according to your comment. And delete mean and sd words, which in line 139 and Table 1.
Comment No.6: Report how were weight and height tested and equipment used.
Feedback: we tested the weight and height of participants with no shoes and Single clothes and trousers on a automatic assessment machine, and the equipment we used was produced by Shenzhen Hengkang JIAYE Technology Co., Ltd, Shenzhen Guangdong province, China(product code: HK6800-ST, http://www.szhkjy.com/). This product was authorized by the Ministry of Education of the PRC.
Comment No.7: Report activity levels for your participants as this can affect their starting level of motor abilities.
Response: Indeed, as you said, the activity level of the participants may affect the experimental results, and we only applied randomization to avoid this risk, such as: we used a randomized controlled experimental design, and the randomization procedure was performed by an independent non-experimental design professional Performed by humans, followed by randomization by computer-generated sequences and randomized using a sealed opaque envelope. Although these works can play a role, there is still much room for improvement, such as assessing children's activity levels through questionnaires, interviews, or accelerometer, which may further improve the experimental design. Due to the impact of the epidemic, we encountered a lot of irresistible troubles during the experiment. Therefore, we also believe that the experiment has room for improvement, and we also make a relevant description of the limitations of this paper. We put forward feasibility suggestions in the section "Study limitation And Future Research" on lines 531 - 536 of the article.
Comment No.8: Motor skills: Line 152 – Please explain why did you use a testing battery that is used for children with motor disorders if your sample of children did not report any disabilities? From this point of view, children with normal development should perform very well in this kind of test and I don’t see a point in the results. Please elaborate
Response: MABC-2 is a common tool for children's motor skill test, and many countries, including China, Japan, etc., have established their own basic norm. It is the gold standard for evaluating children's coordination ability in 3-16. It evaluates children's coordination development level by testing children's hand fine motor (three items), gross motor (two items) and balance ability (three items). A child is at risk for a coordination disorder if the test scores are 5% below the normative level. MABC-2 has been used by a large number of researchers in the study of children's motor skill development. For example:
Samuel W. Logan, Leah E. Robinson, Mary E. Rudisill, Danielle D. Wadsworth & Maria Morera (2014) The comparison of school-age children's performance on two motor assessments: the Test of Gross Motor Development and the Movement Assessment Battery for Children, Physical Education and Sport Pedagogy, 19:1, 48-59.
Ellinoudis, T., Evaggelinou, C., Kourtessis, T., Konstantinidou, Z., Venetsanou, F., & Kambas, A. (2011). Reliability and validity of age band 1 of the movement assessment battery for children – second edition. Research in Developmental Disabilities, 32(3), 1046–1051.
Holm, I., Tveter, A. T., Aulie, V. S., & Stuge, B. (2013). High intra- and interrater chance variation of the movement assessment battery for children 2, ageband 2. Research in Developmental Disabilities, 34(2), 795–800.
Kita, Y., Suzuki, K., Hirata, S., Sakihara, K., Inagaki, M., & Nakai, A. (2016). Applicability of the movement assessment battery for children-second edition to Japanese children: A study of the Age Band 2. Brain and Development, 38(8), 706–713.
Psotta, R., & Brom, O. (2016). Factorial structure of the movement assessment battery for children test-second edition in preschool children. Perceptual and Motor Skills, 123(3), 702–716.
金华,王静,秦志强,柏丹丹,马玉杰,古桂雄. 晚期早产与学龄前期儿童发育性运动协调障碍的相关研究[J].中国儿童保健杂志,2015,23(10):1084-1086.(Relevant study on late-preterm birth and developmental and coordination disorder in preschool children, Chinese Journal of Child Health Care, 2015,23(10):1084-1086.)
花静,吴擢春,古桂雄,孟炜. 儿童运动协调能力成套评估工具的应用性研究[J].中华流行病学杂志,2012(10):1010-1015.(A study on the application of a set of assessment tools for children's motor coordination ability,Chinese Journal of Epidemiology,2012(10):1010-1015)
Comment No.9: How many times was each test performed, what was measured, and which result was taken into further analysis needs to be reported for every test. Was there any demonstration prior to the test? Please explain.
Feedback: The MABC-2 is separated into three sub scales: manual dexterity (three tasks), aiming and catching (two tasks), and balance (three tasks). Manual dexterity was based on the completion times for three activities: placing coins, stringing beads, and drawing paths. Aiming and catching were based on the number of successful hits and catches in two activities: for aiming, participants successively threw 10 bean bags measuring 12×12 cm at a distance of 1.8 m from a target mat; for catching, participants were required to successively catch the same bean bags as thrown from a distance of 1.8 m. Balance was based on three activities: one-legged standing, walking on tiptoe in a straight line, and two-legged continuous jumping. For one-legged standing, participants stood still on their left or right leg, with the respective lengths of time recorded. For walking on tiptoe, participants slightly lifted their heels while walking a straight line measuring 4.5 m in length and 0.25 m in width, with the number of steps recorded. For two-legged continuous jumping(5 times), participants used both legs to jump on six mats of different colors in sequence, with the number of successful attempts recorded. These requires can also be check in the article from line 170 to 187.
Basically, we test twice for each task and take the better score as the test result, and before the test beginning, a trained researcher does a presentation for the participants and allow the participant try once, if he or she made a mistake, correct him or her immediately. Except for aiming and catching, which can be practiced 5 times, other items can be practiced once, and the test can be carried out after the practice. But once the test begins, no response any more. During each session of data collection, approximately ten to twelve children were divided among the six stations. Children rotated from station to station, with the aid of research assistants, until the entire MABC-2 was completed.
Comment No.10:Also report, at what time and where was this done. How many people evaluated this test?
Feedback: The testing time is 8:00-11:30, in a ventilated, bright and quiet gymnasium.
The administration of the MABC-2 was divided into six stations: two stations for manual dexterity, two station for aiming and catching, and two station for balance.
There were eight trained researchers, five faculty members and three graduate students in the area of Motor Behavior who administered the MABC-2. Each researcher had prior experience in administering motor assessments in both research and practical settings. One to two researchers were assigned to each station to ensure accurate scoring of the tasks. All researchers attended a 2 h training session to become familiar with the assessment procedures. Scoring was based on quantitative measures such as the number of successful trials and the length of time taken to complete a task. For each sub scale, raw scores were converted to age-specific standard scores. Standard scores for each sub scale were summed and converted to an age-specific standard and percentile score for total performance on the MABC-2.
We have showed our test teammates and testing site in attachment 2.
Comment No.11:Lines 170-172. Please explain how did you calculate reliability and validity? To which test was this battery compared for validity? Please report. Also, how did you calculate the reliability? From your description, it looks like you calculated it from data before and after the intervention. Also, report exactly how was internal consistency calculated (compared to what)? Elaborate and report
Feedback: we did not calculate reliability and validity of MABC-2, including its internal consistency. As is mentioned before, it is a standardized instrument for measuring motor coordination in children aged 3 to 16 years, it is widely used to screen children with motor coordination and evaluate motor skills in young children.
We refer to and use the reliability and validity of MABC-2 according to the references No. 1 and 2. That was:
Haga, M. The relationship between physical fitness and motor competence in children. Child Care Health Dev 2008, 34, 329-334.
Wuang, Y.; Su, J.; Su, C. Reliability and responsiveness of the Movement Assessment Battery for Children-Second Edition Test in children with developmental coordination disorder. Developmental Medicine & Child Neurology 2012, 54, 160-165.
According to the study results of above, the MABC-2 had a retest reliability of 0.75 and validity of 0.70, with good internal consistency (Cronbach's α range of 0.81-0.90); the total scale and both its sub-items and sub-scales showed good retest reliability. This mentioned in our article from line 206 to 208.
According to Samuel W. Logan, Leah E. Robinson, Mary E. Rudisill, Danielle D. Wadsworth & Maria Morera (2014) (The comparison of school-age children's performance on two motor assessments: the Test of Gross Motor Development and the Movement Assessment Battery for Children, Physical Education and Sport Pedagogy, 19:1, 48-59), MABC-2 had been compared with The Test of Gross Motor Development-2 (TGMD-2). The authors suggest that the MABC-2 assesses motor skills, which is a specific aspect of motor competence.
Comment No.12:Sensory Integration Ability – please report when was this done and who evaluated your participants. Please describe those 5 items of the assessment.
Feedback: Sensory Integration Ability Development Assessment Scale for Children was done twice, once on seven days before the MABC-2 test and once on seven days after MABC-2 test. Printed scales were answered by the guardian (participants’ parents) at home then bring them back to researcher.(see article from line 224 to 226)
We only tested the first three items because the participants we chose were not concerned with insufficient learning ability, and special problems of older age.(see article from line 215 to 217)
There are 14 different questions about the item of vestibular balance dysfunction, which mainly refers to the poor balance of the body, involving the physical movement ability.
such as:
1)Although I saw it, I often collided with tables and chairs, bystanders, columns, doors and walls.
2)move uneasily, touch and gossip, and don't listen to persuasion, the punishment is not invalid.
3)Unable to distinguish between left and right directions, shoes and clothes are often worn upside down.
And there are 21 different questions about the item of tactile over-defense, which mainly refers to emotional instability, hypersensitivity or slow reaction in the process of being touched or touched
such as:
1)be particularly irritable and unreasonable towards his relatives, and afraid of strange environments.
2)Watch TV or listen to stories, easily moved, shout or laugh, afraid of terrorist scenes.
3)Protect his or her own things too much, especially hate others approaching him from behind.
And there are 12 different questions about the item of proprioceptive dysfunction, which mainly refers to poor body coordination, poor perception of the spatial position of various parts of the body, and uncoordinated body movements.
such as:
1)Unclear language, poor pronunciation, slow development of language ability
2)In kindergarten, I still can't wash my hands, wipe my face, cut paper and wipe my ass by myself.
3)It's easy to get lost in a new strange environment.
Comment No.13: Intervention – in this state your study is not reproducible. If could reproduce only the FA group with free activity. However, you need to report, how many times per week was this performed, and how long were these sessions.
Feedback: we performed twice a week in 10 continuous weeks for the participants and 30 minutes every time. We put this information in line 253.
Comment No.14: Also, which Chinese Martial arts or styles were trained? Please report
Feedback: The content of martial arts belongs to the style of Changquan. The criteria for content selection include: coordination of upper and lower limbs, coordination of limbs and body, coordination of visual movements and hand movements. These contents were edited by the State Sports General Administration of PRC. We put the details in attachment 1.
Comment No.15: How can I, from your description reproduce the MAST and MATT training? I can’t!
Therefore, please provide supplemental material describing exact elements that were thought and sequences with photo material, preferably! Also, provide information on what was thaught by weeks.
Feedback: we did a supplemental material attached by the letter, the contents include Illustration of teaching intervention contents, Weekly teaching schedule, Teaching case and Pictures of records of teaching moment. Please check the details in attachment 1.
Comment No.16: Statistical methods – how was the normality of data tested? Please report
Response: We first applied the Shapiro-Wilk test to test whether the data conformed to the normal distribution, and then combined the histogram, P-P and Q-Q plots to verify the data results. This part of the content has been supplemented in the paper. Modifications are distributed in lines 274-275.
Thank you again for your positive and constructive comments and suggestions.
Best regard!

Reviewer 2 Report
The main aim of this study was to examine the effects of Chinese martial arts on the motor skills of preschool children aged 5-6 years through a randomized controlled trial. Regarding the authors, I would like to congratulate and thank them for their effort and motivation involved in this research study. The presentation of the research is well documented, with a scientific basis and respects the latest standards regarding the highest level scientific publications. The methodology was chosen correctly. The conclusions support and result from the research and open new directions for future research. The submitted work is interesting and essentially exhausts the subject under discussion.
Basically, I have no objections. Please adapt the tables in the manuscript to the templates provided in the MDPI Instruction for Authors (also in terms of font, escpecially tables 1 and 2). Please also correct the bibliography, which is currently duplicated in some places with the same footnotes as before (see, for example, footnotes 10 and 11 or 18 and 19, although there are more such examples). The bibliography is worth reinforcing with recent articles included in the journals Archives of Budo and Archives of Budo Science of Martial Arts and Extreme Sports, which essentially covers martial arts.
Supplementing the article with the above-mentioned issues is the only point that I think needs to be added. I keep my fingers crossed for the final success of the publication in the International Journal of Environmental Research and Public Health, which in my opinion is really good.
Author Response
Dear reviewer,
On behalf of my co-authors, we are very grateful to you for giving us an opportunity to revise our manuscript. We appreciate you very much for your positive and constructive comments and suggestions on our manuscript entitled “Effects of Chinese Martial Arts on Motor Skills in Children Between 5 and 6 Years of Age: A Randomized Controlled Trial”.
We have studied reviewers’ comments carefully and tried our best to revise our manuscript. According to your comments, we have already adapt the tables in the manuscript to the templates provided in the MDPI Instruction for Authors. And we have corrected the bibliography in right orders.
In the preface, we added the content of judo and karate to further explain the influence of martial arts on children's motor skill (because martial arts may gradually develop by improving children's physical fitness, such as strength, reaction speed, endurance, flexibility, etc. Gross and fine motor movements in children). We marked in red line from 59 to 71 in the revised article.
In the discussion part, we also inserted the content of judo and karate, because judo and karate can effectively develop children's leg strength and flexibility, and strength and flexibility are one of the factors that affect the balance of the body, so in the discussion part above content was added in the thesis to demonstrate the influence of Chinese martial arts on children's balance ability. Supplements are also marked in red from line 430 to 432.
Thank you again for your positive and constructive comments and suggestions.
Best regard!
Reviewer 3 Report
GENERAL COMMENT. The study was aimed at studying the effect of a Chines Arts protocol modified including a seosory part (MAST) in a great group of 5-6 years old children, in comparison to a traditional approach.
The results obtained showed a significant difference and impact of the modified approach, and although the effect of chinese martial arts are quite well known, the choice of this population and, specifically, the application of this methd to this popilation is of interest and potentially promising.
Overall, it is a nice and well written paper, I only have a few minor comments and suggestions that are reported below for authors’ sake.
MINOR COMMENTS. According to the results obtained, it seems clear that the MAST approach that provides for sensory teaching has proved more effective. It would therefore be interesting and useful to have a more detailed description of the specific exercises that probably led to this result.
Just to give an example: the data related to balance maintainance showed a significant improvement following the approach with the MAST method, but it is not clear which exercises were likely to have led to this improvement. A similar reasoning can be made with regard to proprioception and other parameters that have improved following the proposed program. Including a table showing a list of the proposed exercises would add value and relevance to the study.
It would be interesting to extend this experimentation to populations with specific balance impairment and learning disorders. This aspect could be highlighted in the paragraph dedicated to future perspectives.
Author Response
Dear reviewer,
On behalf of my co-authors, we are very grateful to you for giving us an opportunity to revise our manuscript. We appreciate you very much for your positive and constructive comments and suggestions on our manuscript entitled “Effects of Chinese Martial Arts on Motor Skills in Children Between 5 and 6 Years of Age: A Randomized Controlled Trial”.
We have studied reviewers’ comments carefully and tried our best to revise our manuscript according to the comments. Then we did a supplemental material attached by the letter, the contents include Illustration of teaching intervention contents, Weekly teaching schedule, teaching case and pictures of records of teaching moment, and so on. Please check the details in attachment.
Thank you again for your positive and constructive comments and suggestions.
Best regard!

Round 2
Reviewer 1 Report
Dear Authors
Thank you for putting effort into addressing the majority of my comments.
However, some parts still need to be corrected for greater text flow.
Firstly the paper requires English proofreading.
Introduction:
Lines 56-70 / The text is hard to read and does not present well the wide usage of combat sports and martial arts in this context. Tray to rewrite it in this manner:
- Multiple research has been carried out in the field of judo where the positive impacts of judo training have been shown to impact increased physical abilities of children when compared to other sports or inactive children (and you add all references).
You can for similar sentences for karate and other combat sports and martial arts.
Line 123 - add reference of G*Power software and not a link - https://pubmed.ncbi.nlm.nih.gov/17695343/
line 129 - who were diagnosed by - correct
Line 137 - which software did you use for randomisation on the computer? Report
Table 1 - add space before brackets (cm) and (kg), also add space before and after ± in results for clarity of results.
line 219 - the recovery rate was - correct
Line 244-245 The new sentence is added in without any context. Please rephrase (The practice lasted for...........)
Line 245 - remove the word interesting as interesting for you might not be for someone else :)
Intervention - you need to mention in the intervention description that detailed elements, sequences and training plans are presented in the supplementary material. Please add this sentence
Additionally, in my opinion, part 4 and attachment 2 with pictures of tests and children's training, are not needed. I suggest deleting them.
Conclusion - it feels like something is missing - What are your recommendations for practice? ; therefore, its use is recommended to be regularly included in the school settings. A sentence like this would be beneficial.
Overall there was a great improvement from the initial version of the paper. Some more corrections are needed but the paper is heading in the right direction.
Kind regards
Author Response
Dear Reviewer,
Thank you so much for your warmhearted help on our article. We learned a lot from your rigorous scholarship. After studying the comments below carefully, we have made in response to the questions and suggestions on an item-by-item basis. We appreciated for the hard work of the editor and reviewer!
Comment No.1:Lines 56-70 / The text is hard to read and does not present well the wide usage of combat sports and martial arts in this context. Tray to rewrite it in this manner:
Response: Thank you very much for your suggestion. Indeed, as you said, the newly added content in the introduction does not explain the topic very well, and thank you especially for teaching me how to use a long sentence to express a complete meaning. This is also what we must strengthen in our writing in the future. According to your suggestion and following the logic of introduction writing, the new content has been revised. Please check in the article from line 56 to 64.
Comment No.2:Line 123 - add reference of G*Power software and not a link - https://pubmed.ncbi.nlm.nih.gov/17695343/
Response: Thank you very much for reminding us that it was not suitable putting a link there and actually it was from one of references we referred to. And we have already deleted it. And added the corresponding references to the manuscript.
Comment No.3:line 129 - who were diagnosed by - correct
Response: we felt a little bit shame and we added the word ‘were’ before the diagnosed.
Comment No.4:Line 137 - which software did you use for randomisation on the computer? Report
Response:We mainly use the RAND function of excel software to randomly number all the subjects and generate random sequences. We have supplemented it in the manuscript.
Comment No.5:Table 1 - add space before brackets (cm) and (kg), also add space before and after ± in results for clarity of results.
Response: we added spaces in table 1 according to your kindly reminder.
Comment No.5:line 219 - the recovery rate was - correct
Response: we have changed the word ’is’ to ’was’.
Comment No.6:Line 244-245 The new sentence is added in without any context. Please rephrase (The practice lasted for...........)
Response: we have completed the sentence according to your recommendation.
Comment No.7:Line 245 - remove the word interesting as interesting for you might not be for someone else :)
Response: we totally agree with you and have deleted the word ‘ interesting’.
Comment No.8:Intervention - you need to mention in the intervention description that detailed elements, sequences and training plans are presented in the supplementary material. Please add this sentence
Additionally, in my opinion, part 4 and attachment 2 with pictures of tests and children's training, are not needed. I suggest deleting them.
Response: we added the sentence in the article from line 245 to 246. And deleted the part 4 and attachment 2 of attachment according to your suggestions.
Comment No.9:Conclusion - it feels like something is missing - What are your recommendations for practice? ; therefore, its use is recommended to be regularly included in the school settings. A sentence like this would be beneficial.
Response:Whether the research results can be popularized and applied is an embodiment of the value of a study. Your suggestion is very useful for us. We have added the corresponding content to the conclusion according to your comment.
About the proofreading things, we did have done that work before the first round of submitting which certified by MDPI(3-7-2022). And the revised part of the article has already been checked for correct using of grammar and common technical terms by a experienced, native English speaking scholar.
Yours sincerely
